# Thy Friend is My Friend: Iterative Collaborative Filtering for Sparse Matrix Estimation

**Christian Borgs**
borgs@microsoft.com

**Jennifer Chayes**
jchayes@microsoft.com

**Christina E. Lee**
celee@mit.edu

Microsoft Research New England
One Memorial Drive, Cambridge MA, 02142

**Devavrat Shah**
devavrat@mit.edu
Massachusetts Institute of Technology
77 Massachusetts Ave, Cambridge, MA 02139

## Abstract

The sparse matrix estimation problem consists of estimating the distribution of an $n \times n$ matrix $Y$, from a sparsely observed single instance of this matrix where the entries of $Y$ are independent random variables. This captures a wide array of problems; special instances include matrix completion in the context of recommendation systems, graphon estimation, and community detection in (mixed membership) stochastic block models. Inspired by classical collaborative filtering for recommendation systems, we propose a novel iterative, collaborative filtering-style algorithm for matrix estimation in this generic setting. We show that the mean squared error (MSE) of our estimator converges to 0 at the rate of $O(d^2(pn)^{-2/5})$ as long as $\omega(d^5 n)$ random entries from a total of $n^2$ entries of $Y$ are observed (uniformly sampled), $\mathbb{E}[Y]$ has rank $d$, and the entries of $Y$ have bounded support. The maximum squared error across all entries converges to 0 with high probability as long as we observe a little more, $\Omega(d^5 n \ln^5(n))$ entries. Our results are the best known sample complexity results in this generality.

## 1 Introduction

In this work, we propose and analyze an iterative similarity-based collaborative filtering algorithm for the sparse matrix completion problem with noisily observed entries. As a prototype for such a problem, consider a noisy observation of a social network where observed interactions are signals of true underlying connections. We might want to predict the probability that two users would choose to connect if recommended by the platform, e.g. LinkedIn. As a second example, consider a recommendation system where we observe movie ratings provided by users, and we may want to predict the probability distribution over ratings for specific movie-user pairs. The classical collaborative filtering approach is to compute similarities between pairs of users by comparing their commonly rated movies. For a social network, similarities between users would be computed by comparing their sets of friends. We will be particularly interested in the very sparse case where most pairs of users have no common friends, or most pairs of users have no commonly rated movies; thus there is insufficient data to compute the traditional similarity metrics.

To overcome this limitation, we propose a novel algorithm which computes similarities iteratively, incorporating information within a larger radius neighborhood. Whereas traditional collaborative filtering learns the preferences of a user through the ratings of her/his "friends", i.e. users who share similar ratings on commonly rated movies, our algorithm learns about a user through the ratings of

the friends of her/his friends, i.e. users who may be connected through an indirect path in the data. For a social network, this intuition translates to computing similarities of two users by comparing the boundary of larger radius neighborhoods of their connections in the network. While an actual implementation of our algorithm will benefit from modifications to make it practical, we believe that our *approach* is very practical; indeed, we plan to implement it in a corporate setting. Like all such nearest-neighbor style algorithms, our algorithm can be accelerated and scaled to large datasets in practice by using a parallel implementation via an approximate nearest neighbor data structure. In this paper, however, our goal is to describe the basic setting and concept of the algorithm, and provide clear mathematical foundation and analysis. The theoretical results indicate that this method achieves consistency (i.e. guaranteed convergence to the correct solution) for very sparse datasets for a reasonably general Latent Variable Model with bounded entries.

The problems discussed above can be mathematically formulated as a matrix estimation problem, where we observe a sparse subset of entries in an $m \times n$ random matrix $Y$, and we wish to complete or de-noise the matrix by estimating the probability distribution of $Y_{ij}$ for all $(i, j)$. Suppose that $Y_{ij}$ is categorical, taking values in $[k]$ according to some unknown distribution. The task of estimating the distribution of $Y_{ij}$ can be reduced to $k-1$ smaller tasks of estimating the expectation of a binary data matrix, e.g. $Y^t$ where $Y_{ij}^t = \mathbb{I}(Y_{ij} = t)$ and $\mathbb{E}[Y_{ij}^t] = \mathbb{P}(Y_{ij} = t)$. If the matrix that we would like to learn is asymmetric, we can transform it to an equivalent symmetric model by defining a new data matrix $Y' = \begin{bmatrix} 0 & Y \\ Y^T & 0 \end{bmatrix}$. Therefore, for the remainder of the paper, we will assume a $n \times n$ symmetric matrix which takes values in $[0, 1]$ (real-valued or binary), but as argued above, our results apply more broadly to categorical-valued asymmetric matrices. We assume that the data is generated from a Latent Variable Model in which latent variables $\theta_1, \ldots, \theta_n$ are sampled independently from $U[0, 1]$, and the distribution of $Y_{ij}$ is such that $\mathbb{E}[Y_{ij}|\theta_i, \theta_j] = f(\theta_i, \theta_j) \equiv F_{ij}$ for some latent function $f$. Our goal is to estimate the matrix $F$. It is worth remarking that the Latent Variable Model is a canonical representation for exchangeable arrays as shown by Aldous and Hoover [5, 25, 7].

We present a novel algorithm for estimating $F = [F_{ij}]$ from a sparsely sampled dataset $\{Y_{ij}\}_{(i,j)\in\mathcal{E}}$ where $\mathcal{E} \subset [n] \times [n]$ is generated by assuming each entry is observed independently with probability $p$. We require that the latent function $f$ when regarded as an integral operator has finite spectrum with rank $d$. We prove that the mean squared error (MSE) of our estimates converges to zero at a rate of $O(d^2(pn)^{-2/5})$ as long as the sparsity $p = \omega(d^5 n^{-1})$ (i.e. $\omega(d^5 n)$ total observations). In addition, with high probability, the maximum squared error converges to zero at a rate of $O(d^2(pn)^{-2/5})$ as long as the sparsity $p = \Omega(d^5 n^{-1} \ln^5(n))$. Our analysis applies to a generic noise setting as long as $Y_{ij}$ has bounded support.

Our work takes inspiration from [1, 2, 3], which estimates clusters of the stochastic block model by computing distances from local neighborhoods around vertices. We improve upon their analysis to provide MSE bounds for the general latent variable model with finite spectrum, which includes a larger class of generative models such as mixed membership stochastic block models, while they consider the stochastic block model with non-overlapping communities. We show that our results hold even when the rank $d$ increases with $n$, as long as $d = o((pn)^{1/5})$. As compared to spectral methods such as [28, 39, 20, 19, 18], our analysis handles the general bounded noise model and holds for sparser regimes, only requiring $p = \omega(n^{-1})$.

**Related work.** The matrix estimation problem introduced above includes as specific cases problems from different areas of literature: matrix completion popularized in the context of recommendation systems, graphon estimation arising from the asymptotic theory of graphs, and community detection using the stochastic block model or its generalization known as the mixed membership stochastic block model. The key representative results for each of these are mentioned in Table 1. We discuss the scaling of the sample complexity with respect to $d$ (model complexity, usually rank) and $n$ for polynomial time algorithms, including results for both mean squared error convergence, exact recovery in the noiseless setting, and convergence with high probability in the noisy setting. As can be seen from Table 1, our result provides the best sample complexity with respect to $n$ for the general matrix estimation problem with bounded entries noise model and rank $d$, as the other models either require extra $\log(n)$ factors, or impose additional requirements on the noise model or the expected matrix. Similarly, ours is the best known sample complexity for high probability max-error convergence to 0 for the general rank $d$ bounded entries setting, as other results either assume block constant or noiseless.

Table 1: Sample Complexity of Related Literature grouped in sections according to the following areas —matrix completion, 1-bit matrix completion, stochastic block model, mixed membership stochastic block model, graphon estimation, and our results

| Paper | Sample Complexity | Data/Noise | Expected matrix | Guarantee |
|---|---|---|---|---|
| [27] | $\omega(dn)$ | noiseless | rank $d$ | MSE$\to 0$ |
| [28] | $\Omega(dn\max(\log n, d)), \omega(dn)$ | iid Gaussian | rank $d$ | MSE$\to 0$ |
| [37] | $\omega(dn\log n)$ | iid Gaussian | rank $d$ | MSE$\to 0$ |
| [19] | $\Omega(n\max(d, \log^2 n))$ | iid Gaussian | rank $d$ | MSE$\to 0$ |
| [18] | $\omega(dn\log^6 n)$ | indep bounded | rank $d$ | MSE$\to 0$ |
| [32] | $\Omega(n^{3/2})$ | iid bounded | Lipschitz | MSE$\to 0$ |
| [17] | $\Omega(dn\log^2 n\max(d, \log^4 n))$ | noiseless | rank $d$ | exact recovery |
| [27] | $\Omega(dn\max(d, \log n))$ | noiseless | rank $d$ | exact recovery |
| [39] | $\Omega(dn\log^2 n)$ | noiseless | rank $d$ | exact recovery |
| [19] | $\Omega(n\max(d\log n, \log^2 n, d^2))$ | binary entries | rank $d$ | MSE$\to 0$ |
| [20] | $\Omega(n\max(d, \log n)), \omega(dn)$ | binary entries | rank $d$ | MSE$\to 0$ |
| [1, 3] | $\omega(n)^*$ | binary entries | $d$ blocks | partial recovery |
| [1] | $\Omega(n\log n)^*$ | binary entries | $d$ blocks (SBM) | exact recovery |
| [43] | $\Omega(n\log n)^*$ | binary entries | rank $d$ | MSE$\to 0$ |
| [6] | $\Omega(d^2 n\,\mathrm{polylog}\,n)$ | binary entries | rank $d$ | whp error $\to 0$ |
| [40] | $\Omega(d^2 n)$ | binary entries | rank $d$ | detection |
| [4] | $\Omega(n^2)$ | binary entries | monotone row sum | MSE$\to 0$ |
| [44] | $\Omega(n^2)$ | binary entries | piecewise Lipschitz | MSE$\to 0$ |
| [10] | $\omega(n)$ | binary entries | monotone row sum | MSE$\to 0$ |
| this | $\omega(d^5 n)$ | indep bounded | rank $d$, Lipschitz | MSE$\to 0$ |
| work | $\Omega(d^5 n\log^5 n)$ | indep bounded | rank $d$, Lipschitz | whp error $\to 0$ |

*result does not indicate dependence on $d$.

It is worth comparing our results with the known lower bounds on the sample complexity. For the special case of matrix completion with an additive noise model, i.e. $Y_{ij} = \mathbb{E}[Y_{ij}] + \eta_{ij}$ and $\eta_{ij}$ are i.i.d. zero mean, [16, 20] showed that $\omega(dn)$ samples are needed for a consistent estimator, i.e. MSE convergence to $0$, and [17] showed that $dn\log n$ samples are needed for exact recovery. There is a conjectured computational lower bound for the mixed membership stochastic block model of $d^2 n$ even for detection, which is weaker than MSE going to $0$. Recently, [40] showed a partial result that this computational lower bound holds for algorithms that rely on fitting low-degree polynomials to the observed data. Given that these lower bounds apply to special cases of our setting, it seems that our result is optimal in terms of its dependence on $n$ for MSE convergence as well as high probability (near) exact recovery.

Next we provide a brief overview of prior works reported in the Tables 1. In the context of matrix completion, there has been much progress under the low-rank assumption. Most theoretically founded methods are based on spectral decompositions or minimizing a loss function with respect to spectral constraints [27, 28, 15, 17, 39, 37, 20, 19, 18]. A work that is closely related to ours is by [32]. It proves that a similarity based collaborative filtering-style algorithm provides a consistent estimator for matrix completion under the generic model when the latent function is Lipschitz, not just low rank; however, it requires $\tilde{O}(n^{3/2})$ samples. In a sense, ours can be viewed as an algorithmic generalization of [32] that handles the sparse sampling regime and a generic noise model. Most of the results in matrix completion require additive noise models, which do not extend to setting when the observations are binary or quantized. The USVT estimator is able to handle general bounded noise, although it requires a few log factors more in its sample complexity [18]. Our work removes the extra log factors while still allowing for general bounded noise.

There is also a significant amount of literature which looks at the estimation problem when the data matrix is binary, also known as 1-bit matrix completion, stochastic block model (SBM) parameter estimation, or graphon estimation. The latter two terms are found within the context of community

detection and network analysis, as the binary data matrix can alternatively be interpreted as the adjacency matrix of a graph – which are symmetric, by definition. Under the SBM, each vertex is associated to one of $d$ community types, and the probability of an edge is a function of the community types of both endpoints. Estimating the $n \times n$ parameter matrix becomes an instance of matrix estimation. In SBM, the expected matrix is at most rank $d$ due to its block structure. Precise thresholds for cluster detection (better than random) and estimation have been established by [1, 2, 3]. Our work, both algorithmically and technically, draws insight from this sequence of works, extending the analysis to a broader class of generative models through the design of an iterative algorithm, and improving the technical results with precise MSE bounds.

The mixed membership stochastic block model (MMSBM) allows each vertex to be associated to a length $d$ vector, which represents its weighted membership in each of the $d$ communities. The probability of an edge is a function of the weighted community memberships vectors of both endpoints, resulting in an expected matrix with rank at most $d$. Recent work by [40] provides an algorithm for weak detection for MMSBM with sample complexity $d^2 n$, when the community membership vectors are sparse and evenly weighted. They provide partial results to support a conjecture that $d^2 n$ is a computational lower bound, separated by a gap of $d$ from the information theoretic lower bound of $dn$. This gap was first shown in the simpler context of the stochastic block model [21]. [43] proposed a spectral clustering method for inferring the edge label distribution for a network sampled from a generalized stochastic block model. When the expected function has a finite spectrum decomposition, i.e. low rank, then they provide a consistent estimator for the sparse data regime, with $\Omega(n \log n)$ samples.

Graphon estimation extends SBM and MMSBM to the generic Latent Variable Model where the probability of an edge can be any measurable function $f$ of real-valued types (or latent variables) associated to each endpoint. Graphons were first defined as the limiting object of a sequence of large dense graphs [14, 22, 34], with recent work extending the theory to sparse graphs [12, 13, 11, 41]. In the graphon estimation problem, we would like to estimate the function $f$ given an instance of a graph generated from the graphon associated to $f$. [23, 29] provide minimax optimal rates for graphon estimation; however a majority of the proposed estimators are not computable in polynomial time, since they require optimizing over an exponentially large space (e.g. least squares or maximum likelihood) [42, 10, 9, 23, 29]. [10] provided a polynomial time method based on degree sorting in the special case when the expected degree function is monotonic. To our knowledge, existing positive results for sparse graphon estimation require either strong monotonicity assumptions [10], or rank constraints as assumed in the SBM, the 1-bit matrix completion, and in this work.

We call special attention to the similarity based methods which are able to bypass the rank constraints, relying instead on smoothness properties of the latent function $f$ (e.g. Lipschitz) [44, 32]. They hinge upon computing similarities between rows or columns by comparing commonly observed entries. Similarity based methods, also known in the literature as collaborative filtering, have been successfully employed across many large scale industry applications (Netflix, Amazon, Youtube) due to its simplicity and scalability [24, 33, 30, 38]; however the theoretical results have been relatively sparse. These recent results suggest that the practical success of these methods across a variety of applications may be due to its ability to capture local structure. A key limitation of this approach is that it requires a dense dataset with sufficient entries in order to compute similarity metrics, requiring that each pair of rows or columns has a growing number of overlapped observed entries, which does not hold when $p = o(n^{-1/2})$. This work overcomes this limitation in an intuitive and simple way; rather than only considering directly overlapped entries, we consider longer "paths" of data associated to each row, expanding the set of associated datapoints until there is sufficient overlap. Although we may initially be concerned that this would introduce bias and variance due to the sparse sampling, our analysis shows that in fact the estimate does converge to the true solution.

The idea of comparing vertices by looking at larger radius neighborhoods was introduced in [1], and has connections to belief propagation [21, 3] and the non-backtracking operator [31, 26, 36, 35, 8]. The non-backtracking operator was introduced to overcome the issue of sparsity. For sparse graphs, vertices with high-degree dominate the spectrum, such that the informative components of the spectrum get hidden behind the high degree vertices. The non-backtracking operator avoids paths that immediately return to the previously visited vertex in a similar manner as belief propagation, and its spectrum has been shown to be more well-behaved, perhaps adjusting for the high degree vertices, which get visited very often by paths in the graph. In our algorithm, the neighborhood paths are defined by first selecting a rooted tree at each vertex, thus enforcing that each vertex along a path

in the tree is unique. This is important in our analysis, as it guarantees that the distribution of vertices at the boundary of each subsequent depth of the neighborhood is unbiased, since the sampled vertices are freshly visited.

## 2 Model

We shall use graph and matrix notations in an interchangeable manner. For each pair of vertices (i.e. row or column indices) $u, v \in [n]$, let $Y_{uv} \in [0, 1]$ denote its random realization. Let $\mathcal{E}$ denote the edges. If $(u, v) \in \mathcal{E}$, $Y_{uv}$ is observed; otherwise it is unknown.

- Each vertex $u \in [n]$ is associated to a latent variable $\theta_u \sim U[0, 1]$ sampled i.i.d.
- For each $(u, v) \in [n] \times [n]$, $Y_{uv} = Y_{vu} \in [0, 1]$ is a bounded random variable. Conditioned on $\{\theta_i\}_{i \in [n]}$, the random variables $\{Y_{uv}\}_{1 \leq u < v \leq n}$ are independent.
- $F_{uv} := \mathbb{E}\left[Y_{uv} \mid \{\theta_w\}_{w \in [n]}\right] = f(\theta_u, \theta_v) \in [0, 1]$ for a symmetric $L$-Lipschitz function $f$.
- The function $f$, when regarded as an integral operator, has finite spectrum with rank $d$. That is,

$$f(\theta_u, \theta_v) = \sum_{k=1}^{d} \lambda_k q_k(\theta_u) q_k(\theta_v),$$

  where $q_k$ are orthonormal $L_2$-integrable basis functions. We assume that there exists some $B$ such that $|q_k(y)| \leq B$ for all $k$ and $y \in [0, 1]$.
- For every (unordered) index pair $(u, v)$, the entry is observed independently with probability $p$, i.e. $(u, v) \in \mathcal{E}$ and $M_{uv} = M_{vu} = Y_{uv}$. If $(u, v) \notin \mathcal{E}$, then $M_{uv} = 0$.

The data $(\mathcal{E}, M)$ can be viewed as a weighted undirected graph over $n$ vertices with each $(u, v) \in \mathcal{E}$ having weights $M_{uv}$. The goal is to estimate the matrix $F = [F_{uv}]_{u, v \in [n]}$. Let $\Lambda$ denote the $d \times d$ diagonal matrix with $\{\lambda_k\}_{k \in [d]}$ as the diagonal entries. Let the eigenvalues be sorted in such a way that $|\lambda_1| \geq |\lambda_2| \geq \cdots \geq |\lambda_d| > 0$. Let $Q$ denote the $d \times n$ matrix where $Q(k, u) = q_k(\theta_u)$. Since $Q$ is a random matrix depending on the sampled $\theta$, it is not guaranteed to be an orthonormal matrix (even though $q_k$ are orthonormal functions). By definition, it follows that $F = Q^T \Lambda Q$. Let $d'$ be the number of distinct valued eigenvalues. Let $\tilde{\Lambda}$ denote be the $d \times d'$ matrix where $\tilde{\Lambda}(a, b) = \lambda_b^{a-1}$.

**Discussing Assumptions.** The latent variable model imposes a natural and mild assumption, as Aldous and Hoover proved that if the network is exchangeable, i.e. the distribution over edges is invariant under permutations of vertex labels, then the network can be equivalently represented by a latent variable model [5, 25, 7]. Exchangeability is reasonable for anonymized datasets for which the identity of entities can be easily renamed. Our model additionally requires that the function is $L$-Lipschitz and has finite spectrum when regarded as an integral operator, i.e. $F$ is low rank; this includes interesting scenarios such as the mixed membership stochastic block model and finite degree polynomials. We can also relax the condition to piecewise Lipschitz, as we only need to ensure that for every vertex $u$ there are sufficiently many vertices $v$ which are similar in function value to $u$. We assume observations are sampled independently with probability $p$; however, we discuss a possible solution for dealing with non-uniform sampling in Section 5.

## 3 Algorithm

The algorithm that we propose uses the concept of local approximation, first determining which datapoints are similar in value, and then computing neighborhood averages for the final estimate. All similarity-based collaborative filtering methods have the following basic format:

1. Compute distances between pairs of vertices, e.g.,

$$\texttt{dist}(u, a) \approx \int_0^1 (f(\theta_u, t) - f(\theta_a, t))^2 dt. \tag{1}$$

2. Form estimate by averaging over "nearby" datapoints,

$$\hat{F}_{uv} = \frac{1}{|\mathcal{E}_{uv}|} \sum_{(a,b) \in \mathcal{E}_{uv}} M_{ab}, \tag{2}$$

where $\mathcal{E}_{uv} := \{(a, b) \in \mathcal{E} \text{ s.t. } \texttt{dist}(u, a) < \eta_n, \texttt{dist}(v, b) < \eta_n\}$.

The choice of $\eta_n = \Theta(d(c_1 pn)^{-2/5})$ will be small enough to drive the bias to zero, ensuring the included datapoints are close in value, yet large enough to reduce the variance, ensuring $|\mathcal{E}_{uv}|$ diverges.

**Inutition.** Various similarity-based algorithms differ in the distance computation (Step 1). For dense datasets, i.e. $p = \omega(n^{-1/2})$, previous works have proposed and analyzed algorithms which approximate the $L_2$ distance of (1) by using variants of the finite sample approximation,

$$\texttt{dist}(u, a) = \tfrac{1}{|\mathcal{X}_{ua}|} \sum_{y \in \mathcal{X}_{ua}} (F_{uy} - F_{ay})^2, \tag{3}$$

where $y \in \mathcal{X}_{ua}$ iff $(u, y) \in \mathcal{E}$ and $(a, y) \in \mathcal{E}$ [4, 44, 32]. For sparse datasets, with high probability, $\mathcal{X}_{ua} = \emptyset$ for almost all pairs $(u, a)$, such that this distance cannot be computed.

In this paper we are interested in the sparse setting when $p$ is significantly smaller than $n^{-1/2}$, down to the lowest threshold of $p = \omega(n^{-1})$. If we visualize the data via a graph with edge set $\mathcal{E}$, then (3) corresponds to comparing common neighbors of vertices $u$ and $a$. A natural extension when $u$ and $a$ have no common neighbors, is to instead compare the $r$-hop neighbors of $u$ and $a$, i.e. vertices $y$ which are at distance exactly $r$ from both $u$ and $a$. We compare the product of weights along edges in the path from $u$ to $y$ and $a$ to $y$ respectively, which in expectation approximates

$$\int_{[0,1]^{r-1}} f(\theta_u, t_1)(\prod_{s=1}^{r-2} f(t_s, t_{s+1})) f(t_{r-1}, \theta_y) d\vec{t} = \sum_k \lambda_k^r q_k(\theta_u) q_k(\theta_y) = e_u^T Q^T \Lambda^r Q e_y. \tag{4}$$

We choose a large enough $r$ such that there are sufficiently many "common" vertices $y$ which have paths to both $u$ and $a$, guaranteeing that our distance can be computed from a sparse dataset.

**Algorithm Details.** We present and discuss details of each step of the algorithm, which primarily involves computing pairwise distances (or similarities) between vertices.

*Step 1: Sample Splitting.* We partition the datapoints into disjoint sets, which are used in different steps of the computation to minimize correlation across steps for the analysis. Each edge in $\mathcal{E}$ is independently placed into $\mathcal{E}_1$, $\mathcal{E}_2$, or $\mathcal{E}_3$, with probabilities $c_1$, $c_2$, and $1 - c_1 - c_2$ respectively. Matrices $M_1$, $M_2$, and $M_3$ contain information from the subset of the data in $M$ associated to $\mathcal{E}_1, \mathcal{E}_2$, and $\mathcal{E}_3$ respectively. $M_1$ is used to define local neighborhoods of each vertex, $M_2$ is used to compute similarities of these neighborhoods, and $M_3$ is used to average over datapoints for the final estimate in (2).

*Step 2: Expanding the Neighborhood.* We first expand local neighborhoods of radius $r$ around each vertex. Let $\mathcal{S}_{u,s}$ denote the set of vertices which are at distance $s$ from vertex $u$ in the graph defined by edge set $\mathcal{E}_1$. Specifically, $i \in \mathcal{S}_{u,s}$ if the shortest path in $\mathcal{G}_1 = ([n], \mathcal{E}_1)$ from $u$ to $i$ has a length of $s$. Let $\mathcal{T}_u$ denote a breadth-first tree in $\mathcal{G}_1$ rooted at vertex $u$. The breadth-first property ensures that the length of the path from $u$ to $i$ within $\mathcal{T}_u$ is equal to the length of the shortest path from $u$ to $i$ in $\mathcal{G}_1$. If there is more than one valid breadth-first tree rooted at $u$, choose one uniformly at random. Let $N_{u,r} \in [0,1]^n$ denote the following vector with support on the boundary of the $r$-radius neighborhood of vertex $u$ (we also call $N_{u,r}$ the neighborhood boundary):

$$N_{u,r}(i) = \begin{cases} \prod_{(a,b) \in \text{path}_{\mathcal{T}_u}(u,i)} M_1(a,b) & \text{if } i \in \mathcal{S}_{u,r}, \\ 0 & \text{if } i \notin S_{u,r}, \end{cases}$$

where $\text{path}_{\mathcal{T}_u}(u, i)$ denotes the set of edges along the path from $u$ to $i$ in the tree $\mathcal{T}_u$. The sparsity of $N_{u,r}(i)$ is equal to $\mathcal{S}_{u,r}$, and the value of the coordinate $N_{u,r}(i)$ is equal to the product of weights along the path from $u$ to $i$. Let $\tilde{N}_{u,r}$ denote the normalized neighborhood boundary such that $\tilde{N}_{u,r} = N_{u,r}/|\mathcal{S}_{u,r}|$. We will choose radius $r$ to be $r = \frac{6 \ln(1/p)}{8 \ln(c_1 pn)}$.

*Step 3: Computing the distances.* For each vertex, we present two variants for estimating the distance.

1. For each pair $(u, v)$, compute $\texttt{dist}_1(u, v)$ according to

$$\left(\tfrac{1-c_1 p}{c_2 p}\right) \left(\tilde{N}_{u,r} - \tilde{N}_{v,r}\right)^T M_2 \left(\tilde{N}_{u,r+1} - \tilde{N}_{v,r+1}\right).$$

2. For each pair $(u, v)$, compute distance according to

$$\texttt{dist}_2(u,v) = \sum_{i \in [d']} z_i \Delta_{uv}(r, i),$$

where $\Delta_{uv}(r, i)$ is defined as

$$\left(\tfrac{1-c_1 p}{c_2 p}\right)\left(\tilde{N}_{u,r} - \tilde{N}_{v,r}\right)^T M_2\left(\tilde{N}_{u,r+i} - \tilde{N}_{v,r+i}\right),$$

and $z \in \mathbb{R}^{d'}$ is a vector that satisfies $\Lambda^{2r+2}\tilde{\Lambda}^T z = \Lambda^2 \mathbf{1}$. $z$ always exists and is unique because $\tilde{\Lambda}^T$ is a Vandermonde matrix, and $\Lambda^{-2r}\mathbf{1}$ lies within the span of its columns.

Computing $\texttt{dist}_1$ does not require knowledge of the spectrum of $f$. In our analysis we prove that the expected squared error of the estimate computed in (2) using $\texttt{dist}_1$ converges to zero with $n$ for $p = \omega(n^{-1+\epsilon})$ for some $\epsilon > 0$ and constant rank $d$, i.e. $p$ must be polynomially larger than $n^{-1}$. Although computing $\texttt{dist}_2$ requires knowledge of the spectrum of $f$ to determine the vector $z$, the expected squared error of the estimate computed in (2) using $\texttt{dist}_2$ conveges to zero for $p = \omega(n^{-1})$ and constant rank $d$, which includes the sparser settings when $p$ is only larger than $n^{-1}$ by polylogarithmic factors. We also will show the dependence on $d$ allowing for it to grow slowly with $pn$. It seems plausible that the technique employed by [2] could be used to design a modified algorithm which does not need to have prior knowledge of the spectrum. They achieve this for the stochastic block model case by bootstrapping the algorithm with a method which estimates the spectrum first and then computes pairwise distances with the estimated eigenvalues.

*Step 4: Averaging datapoints to produce final estimate.* The estimate $\hat{F}(u, v)$ is computed by averaging over nearby points defined by the distance estimates $\texttt{dist}_1$ (or $\texttt{dist}_2$). Recall that $B \geq 1$ was assumed in the model definition to upper bound $\sup_{y \in [0,1]} |q_k(y)|$.

Let $\mathcal{E}_{uv1}$ denote the set of undirected edges $(a, b)$ such that $(a, b) \in \mathcal{E}_3$ and both $\texttt{dist}_1(u, a)$ and $\texttt{dist}_1(v, b)$ are less than $\eta_1(n) = 33Bd|\lambda_1|^{2r+1}(c_1 pn)^{-2/5}$. The final estimate $\hat{F}(u, v)$ produced by using $\texttt{dist}_1$ is computed by averaging over the undirected edge set $\mathcal{E}_{uv1}$,

$$\hat{F}(u, v) = \frac{1}{|\mathcal{E}_{uv1}|} \sum_{(a,b) \in \mathcal{E}_{uv1}} M_3(a, b). \tag{5}$$

Let $\mathcal{E}_{uv2}$ denote the set of undirected edges $(a, b)$ such that $(a, b) \in \mathcal{E}_3$, and both $\texttt{dist}_2(u, a)$ and $\texttt{dist}_2(v, b)$ are less than $\xi_2(n) = 33Bd|\lambda_1|(c_1 pn)^{-2/5}$. The final estimate $\hat{F}(u, v)$ produced by using $\texttt{dist}_2$ is computed by averaging over the undirected edge set $\mathcal{E}_{uv2}$,

$$\hat{F}(u, v) = \frac{1}{|\mathcal{E}_{uv2}|} \sum_{(a,b) \in \mathcal{E}_{uv2}} M_3(a, b). \tag{6}$$

## 4 Main Results

We prove bounds on the estimation error of our algorithm in terms of the mean squared error (MSE),

$$\text{MSE} := \mathbb{E}\left[\tfrac{1}{n(n-1)} \sum_{u \neq v} (\hat{F}_{uv} - F_{uv})^2\right],$$

which averages the squared error over all edges. It follows from the model that

$$\int_0^1 (f(\theta_u, y) - f(\theta_v, y))^2 dy = \sum_{k=1}^d \lambda_k^2 (q_k(\theta_u) - q_k(\theta_v))^2 = \|\Lambda Q(e_u - e_v)\|_2^2.$$

The key part of the analysis is to show that the computed distances are in fact good estimates of $\|\Lambda Q(e_u - e_v)\|_2^2$. The analysis essentially relies on showing that the neighborhood growth around a vertex behaves according to its expectation, according to some properly defined notion. The radius $r$ must be small enough to guarantee that the growth of the size of the neighborhood boundary is exponential, increasing at a factor of approximately $c_1 pn$. However, if the radius is too small, then the boundaries of the respective neighborhoods of the two chosen vertices would have a small intersection, so that estimating the similarities based on the small intersection of datapoints would

result in high variance. Therefore, the choice of $r$ is critical to the algorithm and analysis. We are able to prove bounds on the squared error when $r$ is chosen to satisfy the following conditions:

$$r + d' \leq \frac{7\ln(1/c_1 p)}{8\ln(9c_1 pn/8)} = \Theta\left(\frac{\ln(1/c_1 p)}{\ln(c_1 pn)}\right), \quad r + \frac{1}{2} \geq \frac{6\ln(1/p)}{8\ln(7|\lambda_d|^2 c_1 pn/8|\lambda_1|)} = \Theta\left(\frac{\ln(1/p)}{\ln(c_1 pn)}\right). \quad (7)$$

The parameter $d'$ denotes the number of distinct valued eigenvalues in the spectrum of $f$, $(\lambda_1 \ldots \lambda_d)$, and determines the number of different radius "measurements" involved in computing $\texttt{dist}_2(u, v)$. Computing $\texttt{dist}_1(u, v)$ only involves a single measurement, thus the left hand side of (7) can be reduced to $r + 1$ instead of $r + d'$. When $p$ is above a threshold, we choose $c_1$ to decrease with $n$ to ensure (7) can be satisfied, sparsifying the edge set $\mathcal{E}_1$ used for expanding the neighborhood around a vertex . When the sample probability is polynomially larger than $n^{-1}$, i.e. $p = n^{-1+\epsilon}$ for some $\epsilon > 0$, these constraints imply that $r$ is a constant with respect to $n$. However, if $p = \tilde{O}(n^{-1})$, we will need $r$ to grow with $n$ according to a rate of $6\ln(1/p)/8\ln(c_1 pn)$.

**Theorem 4.1.** *If* $p = n^{-1+\epsilon}$ *for some* $\epsilon \in (0, \frac{1}{6})$, *with a choice of* $c_1$ *such that* $c_1 pn = \Theta\left(\max(pn, (p^6 n^7)^{\frac{1}{19}})\right)$, *there exists a constant* $r$ *(with respect to* $n$*) which satisfies* (7). *If* $p = \omega(n^{-1} d^5)$ *and* $|\lambda_d| = \omega((c_1 pn)^{-\frac{1}{4}})$, *then the estimate computed using* $\texttt{dist}_1$ *with parameter* $r$ *achieves*

$$\text{MSE} = O\left(\left(\frac{|\lambda_1|}{|\lambda_d|}\right)^{2r} \frac{B^3 d^2 |\lambda_1|}{(c_1 pn)^{2/5}}\right).$$

*If* $p = \omega(n^{-1} d^5 \ln^5(n))$, *with probability greater than* $1 - O\left(d \exp\left(-\frac{(c_1 pn)^{1/5}}{9B^2 d}\right)\right)$, *the estimate satisfies*

$$\|\hat{F} - F\|_{\max} := \max_{i,j} |\hat{F}_{ij} - F_{ij}| = O\left(\left(\frac{|\lambda_1|}{|\lambda_d|}\right)^r \left(\frac{B^3 d^2 |\lambda_1|}{(c_1 pn)^{2/5}}\right)^{1/2}\right).$$

Theorem 4.1 proves that the mean squared error (MSE) of the estimate computed with $\texttt{dist}_1$ is bounded by $O((|\lambda_1|/|\lambda_d|)^{2r} d^2 (c_1 pn)^{-2/5})$. Therefore, our algorithm with $\texttt{dist}_1$ provides a consistent estimate when $r$ is constant with respect to $n$, which occurs for $p = n^{-1+\epsilon}$ for some $\epsilon > 0$. In fact, the reason why the error blows up with a factor of $(|\lambda_1|/|\lambda_d|)^{-2r}$ is because we compute the distance by summing product of weights over paths of length $2r$. From (4), we see that in expectation, when we take the product of edge weights over a path of length $r$ from $u$ to $y$, instead of computing $f(\theta_u, \theta_y) = e_u^T Q\Lambda Q e_y$, the expression concentrates around $e_u^T Q\Lambda^r Q e_y$, which contains extra factors of $\Lambda^{r-1}$. Therefore, by computing over a radius $r$, the calculation in $\texttt{dist}_1$ will approximate $\|\Lambda^{r+1} Q(e_u - e_v)\|_2^2$ rather than our intended $\|\Lambda Q(e_u - e_v)\|_2^2$, thus leading to an error factor of $(|\lambda_1|/|\lambda_d|)^{2r}$. It turns out that $\texttt{dist}_2$ adjusts for this bias, as the multiple measurements $\Delta_{uv}(r, i)$ with different length paths allows us to separate out $e_k \Lambda Q(e_u - e_v)$ for all $k$ with distinct values of $\lambda_k$.

**Theorem 4.2.** *If* $p = o(n^{-5/6})$, *with a choice of* $c_1$ *such that* $c_1 pn = \Theta\left(\max(pn, (p^6 n^7)^{\frac{1}{(8d'+11)}})\right)$, *there exists a value for* $r$ *which satisfies* (7). *If* $p = \omega(n^{-1} d^5)$, $|\lambda_d| = \omega((c_1 pn)^{-\frac{1}{4}})$, *and* $d = o(r)$, *then the estimate computed using* $\texttt{dist}_2$ *with parameter* $r$ *achieves*

$$\text{MSE} = O\left(\frac{B^3 d^2 |\lambda_1|}{(c_1 pn)^{2/5}}\right).$$

*If* $p = \omega(n^{-1} d^5 \ln^5(n))$, *with probability* $1 - O\left(d \exp\left(-\frac{(c_1 pn)^{1/5}}{9B^2 d}\right)\right)$, *the estimate satisfies*

$$\|\hat{F} - F\|_{\max} := \max_{i,j} |\hat{F}_{ij} - F_{ij}| = O\left(\left(\frac{B^3 d^2 |\lambda_1|}{(c_1 pn)^{2/5}}\right)^{1/2}\right).$$

Theorem 4.2 proves that the mean squared error (MSE) of the estimate computed using $\texttt{dist}_2$ is bounded by $O(d^2 (c_1 pn)^{-2/5})$; and thus the estimate is consistent in the ultra sparse sampling regime of $p = \omega(d^5 n^{-1})$.

# 5 Discussion

In this work we presented a similarity based collaborative filtering algorithm which is provably consistent in sparse sampling regimes, as long as the sample probability $p = \omega(n^{-1})$. The algorithm computes similarity between two users by comparing their local neighborhoods. Our model assumes that the data matrix is generated according to a latent variable model, in which the weight on an observed edge $(u, v)$ is equal in expectation to a function $f$ over associated latent variables $\theta_u$ and $\theta_v$. We presented two variants for computing similarities (or distances) between vertices. Computing $\texttt{dist}_1$ does not require knowledge of the spectrum of $f$, but the estimate requires $p$ to be polynomially larger than $n$ in order to guarantee the expected squared error converges to zero. Computing $\texttt{dist}_2$ uses the knowledge of the spectrum of $f$, but it provides an estimate that is provably consistent for a significantly sparse regime, only requiring that $p = \omega(n^{-1})$. The mean squared error of both algorithms is bounded by $O((pn)^{-1/5})$. Since the computation is based on of comparing local neighborhoods within the graph, the algorithm can be easily implemented for large scale datasets where the data may be stored in a distributed fashion optimized for local graph computations.

**Practical implementation.** In practice, we do not know the model parameters, and we would use cross validation to tune the radius $r$ and threshold $\eta_n$. If $r$ is either too small or too large, then the vector $N_{u,r}$ will be too sparse. The threshold $\eta_n$ trades off between bias and variance of the final estimate. Since we do not know the spectrum, $\texttt{dist}_1$ may be easier to compute, and still enjoys good properties as long as $r$ is not too large. When the sampled observations are not uniform across entries, the algorithm may require more modifications to properly normalize for high degree hub vertices, as the optimal choice of $r$ may differ depending on the local sparsity. The key computational step of our algorithm involves comparing the expanded local neighborhoods of pairs of vertices to find the "nearest neighbors". The local neighborhoods can be computed in parallel, as they are independent computations. Furthermore, the local neighborhood computations are suitable for systems in which the data is distributed across different machines in a way that optimizes local neighborhood queries. The most expensive part of our algorithm involves computing similarities for all pairs of vertices in order to determine the set of nearest neighbors. However, it would be possible to use approximate nearest neighbor techniques to greatly reduce the computation such that approximate nearest neighbor sets could be computed with significantly fewer than $n^2$ pairwise comparisons.

**Non-uniform sampling.** In reality, the probability that entries are observed is not be uniform across all pairs $(i, j)$. However, we believe that an extension of our result can also handle variations in the sample probability as long as the sample probability is a function of the latent variables and scales in the same way with respect to $n$ across all entries. Suppose that the probability of observing $(i, j)$ is given by $pg(\theta_i, \theta_j)$, where $p$ is the scaling factor (contains the dependence upon $n$), and $g$ allows for constant factor variations in the sample probability across entries as a function of the latent variables. If we let matrix $X$ indicate the presence of an observation or not, then we can apply our algorithm twice, first on matrix $X$ to estimate function $g$, and then on data matrix $M$ to estimate $f$ times $g$. We can simply divide by the estimate for $g$ to obtain the estimate for $f$. The limitation is that if $g(\theta_i, \theta_j)$ is very small, then the error in estimating the corresponding $f(\theta_i, \theta_j)$ will have higher variance. However, it is expected that error increases for edge types with fewer samples.

### Acknowledgments

This work is supported in parts by NSF under grants CMMI-1462158 and CMMI-1634259, by DARPA under grant W911NF-16-1-0551, and additionally by a NSF Graduate Fellowship and Claude E. Shannon Research Assistantship.

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
