[Reviews · NeurIPS 2017]

Reviewer 1



The paper is about link prediction in graphs. A highly relevant problem in graph / network analysis problems that arise in recommender systems and drug-protein interaction prediction for example. The paper is extremely well written. It was a pleasure for me to read the very well documented state of the art and related work section. I was able to find only a single typo: l.210 r-top and not r-hop. Technically the paper's models and contributions are involved. The authors use graphon models which are limit objects of sequences of graphs and propose to study a nearest neighbors scheme under the model that is a sequence of graphs with n nodes where n-- > infinity but with a fixed sparsity level p = omega( d^2 / n), where d is the rank of the underlying latent factor space. The authors prove that the proposed approach 's Mean squared error goes to zero when omega(d^2 n) entries are observed and the Max squared error goes to zero when Omega (d^2 n log n) entries are observed. Compared to existing results these results remove a log factor, while relaxing the assumption on the noise. The paper uses a graph that has weighted edges, not a standard binary graph. The model is not entirely new. The algorithm probably requires a more compact presentation. The algorithm is presented in pages 5 and 6 and some of the notations are a little confusing: it took me time to find the definitions of M,M_1,M_2,M_3

Reviewer 2



This paper studies a rather generic form of sparse matrix estimation problem. Inspired by [1-3], it suggests an algorithm that is roughly based on averaging matrix elements that are r-hop common neighbours of the two of nodes for which the estimate is being computed. The authors then prove sample complexity bounds for the MSE to go to zero. While there is a lot of related work in this direction the present paper discussed this nicely and treats the many cases in a unified framework and allows for generic noise function and sparser matrices. The paper provides (as far as I know) best bounds that hold in this generality. I think this is a very good paper that should be accepted. Some comments suggesting improvements follow: While in the sparse regime treating the generality of the noise is interesting and as far as I know original. It might be worth mentioning that in the denser case results on universality with respect to the noise are known for the matrix estimation problem ("MMSE of probabilistic low-rank matrix estimation: Universality with respect to the output channel" by Lesieur et al, and "Asymptotic mutual information for the two-groups stochastic block model" by Deshpande, Abbe, Montanari). The algorithm seems related to a spectral algorithms designed for sparse graphs, based on the non-backtracking matrix. While the introduction mentions in only one sentence relation to only standard spectral algorithm, the relation to belief propagation and non-backtracking operator is discussed on page 6. At the same time the original works suggesting and analyzing those approaches are not cited there (i.e. "Spectral redemption in clustering sparse networks" by Krzakala et al, or "Non-backtracking spectrum of random graphs: community detection and non-regular Ramanujan graphs" by Bordenave et al.). This relation should perhaps be discussed in the section of related work or in the introduction and not in the section describing in details of the algorithm. The introduction discusses an unpublished work on mixed membership SBM, in particular a gap between information theoretical lower bound and a conjectures algorithmic threshold in that case. Perhaps it should be mentioned that such results originate from the normal SBM where both the information-theoretic threshold for detection, and the conjectured algorithmic threshold were studied in detail, e.g. in "Asymptotic analysis of the stochastic block model for modular networks and its algorithmic applications" by Decelle et al. Also in that case the gap between the two threshold is d (for large d). While the main contribution of the paper is theoretical, it would have been nice to see some practical demonstration of the algorithm, comparison to other algorithms (at the same time this should not be used as an argument for rejection). Evidence of the scalability of the algorithm should be presented. Minor points: While the o(), O(), \Omega() notations are rather standard I was not very familiar with the \omega() and had to look it up to be sure. Perhaps more of NIPS audience would not be familiar with those and the definition could be shortly reminded. I've read the author's feedback and took it into account in my score.

Reviewer 3



This paper proposed an improved iterative algorithm for filling missing entries in a very sparse matrix, typically known as matrix completion and has wide applications in link prediction and recommendation systems. The main ides is to enlarge the choice of the neighborhood samples in case two nodes in the graph do not have a direct linkage, (for example, two nodes can be connected by a series of intermediate nodes on a path) so that the resultant estimation is expected to converge faster. It is shown that the mean squared error of estimation converges to zeros given sufficient samples of O(d^2nlog(n)). No empirical evaluations nor comparisons are provided to support the theoretical analysis. I have several concerns with the proposed method. First, as the authors summarized in Section 3, that existing methods for local approximation typically use two steps, namely, distance computation and averaging. To me such algorithms are quite unstable and depends heavily on the distance measure, while the ground truth distance is usually unknown and hard to estimate if the matrix/graph is very sparse. In (3), the distance between two nodes are expressed by the averaged difference of their neighbors's values. Therefore the distance measure and the neighborhood depends heavily on each other, which are hard to optimize. The theoretic analysis of the paper does not show clearly why the proposed scheme leads to a higher convergence rate. In particular while it may be helpful to increase the radius of the neighborhood size, what is a qualitative criteria to avoid too large neighborhood? Also, there lacks empirical results demonstrating the usefulness of the proposed method, therefore it is difficult to judge the practical applicability of the method. It is highly preferred that more experiments and comparisons be provided to make this a solid work.